# NMR-Based Metabolomics Reveals Position-Specific Signatures Associated with Physical Demands in Professional Soccer Players

**DOI:** 10.3390/biomedicines13112583

**Published:** 2025-10-22

**Authors:** Suewellyn N. dos Santos, Glydiston E. O. Ananias, Edmilson R. da Rocha, Alessandre C. Carmo, Edson de S. Bento, Thiago M. de Aquino, Ronaldo V. Thomatieli-Santos, Luiz Rodrigo A. de Lima, Pedro Balikian, Natália de A. Rodrigues, Gustavo G. de Araujo, Filipe A. B. Sousa

**Affiliations:** 1Post-Graduate Program in Health Sciences, Federal University of Alagoas, Maceió 57072-900, AL, Brazil; suewellyn.nunes@hotmail.com (S.N.d.S.); luiz.lima@iefe.ufal.br (L.R.A.d.L.); gustavo.araujo@iefe.ufal.br (G.G.d.A.); 2Post-Graduate Program in Nutrition, Federal University of Alagoas, Maceió 57072-900, AL, Brazil; glydiston@gmail.com; 3Nucleus of Analysis and Research in Nuclear Magnetic Resonance, Institute of Chemistry and Biotechnology, Federal University of Alagoas, Maceió 57072-900, AL, Brazil; edmilsonrrj@gmail.com (E.R.d.R.J.); alessandre89@gmail.com (A.C.C.); edson.bento@iqb.ufal.br (E.d.S.B.); thiago.aquino@iqb.ufal.br (T.M.d.A.); 4Department of Biosciences, Institute of Health and Society, Federal University of São Paulo, Santos 11015-020, SP, Brazil; ronaldo.thomatieli@unifesp.br; 5Research Laboratory in Biodynamics of Human Performance and Health, Institute of Physical Education and Sports, Federal University of Alagoas, Maceió 57072-900, AL, Brazil; 6Laboratory of Applied Sports Science, Institute of Physical Education and Sports, Federal University of Alagoas, Maceió 57072-900, AL, Brazil; pedro.junior@iefe.ufal.br (P.B.J.); natalia.rodrigues@iefe.ufal.br (N.d.A.R.)

**Keywords:** metabolomics, soccer, training, load, metabolism, position

## Abstract

**Background:** Soccer’s varied physical demands require meticulous load monitoring, which is now being advanced by combining GPS for external metrics and NMR-based metabolomics for internal metabolic profiling. This study aimed to investigate how player position influences the metabolomic profile (as a marker of internal load) under known match effort (external load). **Methods:** This was a longitudinal observational descriptive study involving 12 professional soccer players from the U-20 São Paulo Football Club, enrolled in the 2022 São Paulo State Under-20 Football Championship. Players were monitored across six matches during the season, culminating in a total of 49 individual match observations from those players (4-2-3-1 formation: Central Defenders [CD], *n* = 9; Full Backs [FB], *n* = 9; Central Midfielders [CM], *n* = 14; Wide Midfielders [WM], *n* = 12; Forwards [F], *n* = 5). Internal load was assessed via urinary metabolomics, with urine samples collected 24 h post-match. A non-targeted, global metabolomics approach was employed using nuclear magnetic resonance (NMR) spectroscopy. External load was monitored using GPS tracking devices. Multivariate analyses included partial least squares discriminant analysis (PLS-DA), and heat maps. **Results:** Metabolomic analysis identified 38 metabolites with a Variable Importance in Projection (VIP) score > 1.0, revealing perturbations in carbohydrate metabolism and the tricarboxylic acid (TCA) cycle, amino acid and peptide metabolism, pyrimidine metabolism, and ketone body pathways, and effectively discriminating post-match recovery metabolic profiles. External load metrics varied significantly by player position: CMs covered greater distances below 20 km/h (8702.93 ± 1271.89 m), exhibited higher relative distance (114.29 ± 7.67 m/min), total distance (9193.21 ± 1261.35 m), and player load (945.71 ± 135.82 a.u.); CDs achieved higher peak speeds (31.78 ± 1.20 m/s); and WMs performed greater sprint distances (168.11 ± 91.69 m). Metabolomic profiles indicated that CMs showed stronger associations with markers of muscle damage and inflammation, whereas CDs and WMs were more closely linked to energy metabolism and oxidative stress. **Conclusions:** These results highlight the importance of a personalized approach to training load monitoring and recovery strategies, considering the distinct physiological and metabolic demands associated with each player position.

## 1. Introduction

Contemporary technologies now enable the estimation and exploration of mechanical impacts during exercise [1]. In the context of physical performance analysis, these variables are collected via electronic performance tracking systems (EPTS), including global positioning system (GPS) devices, microelectromechanical systems, and computerized video systems [2]. These systems allow for the measurement of external load data—such as time-motion analysis, total distance covered, high-speed running distance, player load, and accelerations and decelerations produced by the athlete [3]. Soccer is a sport characterized by intermittent high-intensity efforts interspersed with longer periods of low-intensity activity during matches [4]. However, the physical demands imposed on players vary based on their playing position, specific role, team formation, match location (home or away), the player’s category, score dynamics, playing style, and the phase of the match [5,6].

For instance, during a match, midfielders cover a greater total distance and perform more high-intensity work than defenders and forwards [7]. According to Pettersen and Brenn [8], midfielders exhibited the highest values for high intensity running (1044.2 m), total running (224.4 m), and accelerations (*n* = 185.2), while central defenders (CD) recorded the lowest values (508.3 m, 85.1 m, and *n* = 119.0, respectively). Wide midfielders (WMs) achieved the highest maximum speed (30.3 km/h), and CD the lowest (28.6 km/h). Regarding soccer athletes playing position, it is important to note, however, that numerous definitions and positional classifications exist in the literature without a clear standard. Generally, a commonly accepted nomenclature states that CD, full-backs (FB), and goalkeepers (GK) operate in the defensive sector; central midfielders (CM), WM, and defensive midfielders (DM) comprise the midfield sector; and forwards (F) act in the attacking or offensive sector [9]. Professional soccer players cover between 9 and 14 km during a match, with high-intensity running accounting for approximately 5–15% of this total distance [10].

Understanding the impact of mechanical loads on players’ physiological responses is fundamental for controlling and appropriately prescribing training. In this context, internal load—which reflects the biological and perceptual responses of the organism to effort—has emerged as an essential marker for determining the adaptations induced by physical stimuli [11]. Given that these responses vary both between individuals and within collective contexts, various strategies have been employed to monitor them. Among the most widely used tools are objective methods, including heart rate, blood lactate levels, oxygen consumption, and the training impulse (TRIMP), as well as subjective measures such as the rating of perceived exertion (RPE), well-being questionnaires, and psychological inventories [3].

Therefore, with the advancement of technologies applied to sports, metabolomics has emerged as a promising approach for investigating exercise physiology and its associated metabolic processes with greater depth and resolution [12]. Metabolomics, as an ‘omics’ discipline devoted to the comprehensive study of biological systems, characterizes metabolites produced and released at systemic and cellular levels in response to physiological stimuli such as physical exercise. These metabolites can be quantified in accessible biological fluids, including blood, saliva, and urine, enabling dynamic metabolic profiling [13,14]. In this context, NMR-based metabolomics represents an innovative tool that bridges fundamental science and performance, supporting precise, preventive, and personalized strategies for monitoring and optimizing athlete health and performance [15,16].

The applicability of metabolomics in soccer has been explored to identify metabolic alterations in athletes induced by training intensity [1]. In an observational longitudinal study, Quintas et al. [13] found an association between external load and the urinary metabolic profile, which included steroid hormones, hypoxanthine metabolites, acetylated amino acids, intermediates in phenylalanine and tyrosine metabolism, tryptophan metabolites, and riboflavin. These alterations were linked to changes in biochemical pathways associated with long-term training adaptation. Given the previously detected differences in external load between playing positions in soccer, along with metabolomics’ capacity to correlate with internal loads, it is plausible to hypothesize that metabolomics would be sensitive enough to detect metabolic profile changes related to playing position. If confirmed, this hypothesis would expand the applicability of metabolomics in sports, as its investigation could allow for: mapping position-specific physiological demands; identifying sensitive and early biomarkers of position-specific fatigue, stress, or overload; and influencing tactical and technical game decisions.

Therefore, understanding the extent to which metabolomic analysis—when associated with players’ external training load—can differentiate positional game demands in soccer would reinforce its utility as a tool for optimizing training programs to achieve desired physiological adaptations for competition. Consequently, this approach could enhance performance and recovery specific to soccer’s demands while mitigating adverse impacts on player health. Given this context, the present study aims to determine the capacity of metabolomics to identify differences between professional soccer players’ positions, associated with the effects of different external loads following match play. We hypothesized that 24 h post-match urinary metabolomics would discriminate between playing positions, reflecting physical demands measured by external load.

## 2. Materials and Methods

### 2.1. Study Design

This observational longitudinal descriptive study involved no intervention or manipulation of variables by the researchers. The recruitment of volunteers was performed by convenience, considering the high level of athletes involved. The study describes the responses of the phenomenon over a seven-week period (September to October 2022), during which players underwent training sessions and played seven matches in the 3rd and 4th Qualifying Phases and the Semifinal Phase of the 2022 São Paulo State Under-20 Football Championship. All matches from board clearance for data collection until the end of the Championship were considered in this analysis. Additionally, all players followed the same training sessions. For the purposes of this study, all analysis was performed in post-match data. All players involved provided informed consent before engaging in any data collection activity.

### 2.2. Participants

Inclusion criteria were: male sex, age between 18 and 20 years, a professional soccer contract with the participant team, and being eligible for full physical training and playing time (free from injury or physiotherapeutic treatment by the medical department). The team was divided into the following playing positions: CD, FB, CM, WM, and F. We excluded GK from this analysis due to their distinct training loads and specific physical game characteristics compared to outfield players. The team’s starting formation was 4-2-3-1 for all included matches.

All 40 players on the São Paulo Football Club (SPFC) U-20 roster were considered for the study, and twelve were selected. The exclusion criteria were not being amongst the starters or frequent substitutes with significant playing time (≥45 min per match), which were considered part of the championship’s core training group by the technical staff. Those exclusion criteria were necessary considering the study’s aims to relate the match external load and the metabolomic profile (internal load). The average total match time was 97.12 ± 2.53 min. The team’s eleven-month macrocycle initiated in November 2021, with the preseason and first team competition in January 2022 (São Paulo Junior Football Cup), the beginning of the state championship in May, and the national championship in June. In August the team was eliminated from the national championship, and by September 2022 was focused only on the state championship for the data collection period included in this analysis.

The players were housed at the club training facilities for the data collection period, and followed the club’s standardized daily regimen, which covered training, rest, and diet. The SPFC President Laudo Natel Training Center is a large, high-end facility featuring multiple soccer fields, a 1500-seat stadium, dormitories, a large dining hall, and a 4-star hotel used for the 2014 World Cup. For the period of data collection (fall), all players included in this analysis were housed at the club’s facilities and received five daily meals. The team’s nutrition staff regularly assessed the players to ensure that the required macronutrient and micronutrient intake was being achieved. The training center houses REFFIS, a dedicated unit with sports medicine professionals who provide comprehensive care, from injury diagnosis and physiotherapy to individualized training programs. During weekly micro cycles, the athletes were engaged in a minimum of seven training sessions, which included physical, technical, and tactical components, each lasting an average of 70 min. In addition to these team activities, some athletes also engaged in at least one individual strength training session. All players included had at least four years of experience competing at the state level, and at least one year at the national and international levels, with experience in the U20 and the Senior team.

On the day after the match (MD + 1), urine samples were collected in the morning from players who participated for at least 45 min. Collection occurred before any exercise or effort. Additionally, data from game 3 were excluded because the urine collection for metabolomics analysis was conducted 48 h post-match due to team logistics constraints, resulting in six games included. The following were excluded from consideration: players with <45 min played; sporadic substitutes; second roster players and players unwilling to urinate before the training session started. The final sample consisted of 49 individual observations of training load data (external and internal) by playing position (CD = 9; FB = 9; CM = 14; WM = 12; F = 5). These observations were derived from the 12 eligible players across the six season games.

### 2.3. Variables

#### 2.3.1. External Match Training Load

External load data were obtained via GPS during daily training sessions and matches. To determine the external load of activities, all players wore a GPS device (Catapult OPTIMEYE S7^®^, Melbourne, Australia) during all training sessions and matches of the season. The device provided the following external load metrics: Total distance covered (Total Dist; m); Relative distance (Relative Dist; m/min); Player Load (a.u.; individual player training load intensity); High-intensity running distance (Dist > 20 km/h; m; total distance covered above 20 km/h); Sprint distance (Sprints Dist; m; >25.0 km/h); Maximum speed (Max Speed; km/h); Explosive efforts (m; distance covered with acceleration greater than 1.12 m/s^2^); Accelerations (*n*; >2.0 m/s^2^); Decelerations (*n*; <−2.0 m/s^2^); Total distance covered below 20 km/h (Dist < 20; m). At the end of each training session and match, the data collected by the device were transferred to proprietary software developed by the manufacturer (Open Field Console, v2.2.1, Catapult, Melbourne, Australia). This software provided the individual movement metrics produced by the players in a spreadsheet, along with the group average values for each metric analyzed.

#### 2.3.2. Metabolic Analysis of Urine Samples

Urine samples were collected to determine the internal training load and perform urinary metabolomic analyses on the players. Urine was chosen as the biofluid due to its advantages of being simple, non-invasive, and quick to collect, making it useful for detecting metabolic alterations. Data collections began during the Third Qualifying Phase of the 2022 São Paulo Under-20 Football Championship. Over a seven-week period, featuring one match per week, 25 mL urine samples were collected 24 h post-match (i.e., before the first training session following the match, MD + 1) to analyze the metabolic impact and responses of the players to the matches. Samples were collected, immediately frozen at −20 °C, and shipped by air for analysis to the Nucleus of Analysis and Research in Nuclear Magnetic Resonance at the Institute of Chemistry and Biotechnology, Federal University of Alagoas.

An untargeted (global) metabolomics approach was employed using nuclear magnetic resonance (NMR). Analysis was conducted on a Bruker 600 MHz spectrometer (AVANCE III, Bruker BioSpin, Ettlingen, Germany) equipped with a 5 mm PABBO probe at 300 K, using the NOESYGPPR1D pulse sequence. From the 25 mL collected post-match, 1.5 mL aliquots were transferred to individual Eppendorf^®^ tubes and centrifuged at 14,000 rpm (ROTANTA 460R, Hettich Zentrifugen, Tuttlingen, Germany) for 15 min. The supernatant was then transferred to new tubes and stored at −80 °C for subsequent analysis. For the NMR analysis, 500 µL of each sample’s supernatant was transferred to a 5 mm NMR tube. Then, 200 µL of a buffer solution (sodium phosphate, pH = 7.4) containing 100% D_2_O and 1 mM TSP (trimethylsilylpropanoic acid, used as a chemical shift standard) was added to each tube. Water signal suppression was achieved through pre-saturation with the following parameters: NS: 128 (number of scans); D1: 4.00 s (relaxation delay); TD: 64K (data points); SW: 20 ppm (spectral width); O1P: 4.69 ppm (irradiation frequency for water suppression); AQ: 5.11 s (acquisition time).

Spectra were processed using TopSpin^®^ software (v. 3.6.5, Bruker Corporation, Billerica, MA, USA). Metabolite peak identification was performed using Chenomx^®^ NMR Suite software (v. 11, Chenomx Inc., Edmonton, AB, Canada) and confirmed against the Human Metabolome Database (HMDB; http://www.hmdb.ca/, accessed on 1 March 2023). Spectral pre-processing was conducted in R software (v. 4.2.2, R Foundation for Statistical Computing, Vienna, Austria) using the PepsNMR package (v. 3.17) and involved procedures for overlapping, region alignment, peak picking, and quantification of spectra. The resulting data were then transferred to Excel^®^ (Microsoft Corporation, Redmond, WA, USA), to form a matrix with samples in rows and the 38 identified metabolites in columns.

### 2.4. Statistical Analysis

Data are presented as median or mean and standard deviation (±). All analyses were conducted using RStudio software, version 13. A one-way ANOVA was used to assess the effect of player position on external training load data and the metabolic profile. When a significant main effect of playing position was found, Tukey’s post hoc tests were performed to identify specific differences between positions. Statistical significance was set at *p* < 0.05.

For metabolomic analysis and GPS-derived metrics, multivariate statistical methods were employed. Discriminant Analysis by Partial Least Squares (PLS-DA) was performed, generating score and loading plots. These plots provide a Variable Importance in Projection (VIP) score. We defined relevant variables as follows: for GPS-derived metrics, a VIP score > 0.8 was considered significant; for urinary metabolites, a more stringent threshold of VIP > 1.0 was applied due to the higher number of variables. Furthermore, integrative analyses were conducted using Pearson correlation plots to examine relationships between the 38 identified metabolites and the external load (GPS) variables obtained from the athletes after the six championship matches. The methodological design of this study is outlined in Figure 1.

## 3. Results

### 3.1. Description of External Training Load by Playing Position

Comparison of external load data obtained via GPS revealed differences in training load metrics across playing positions. The CM group presented the highest values for several metrics. They covered a significantly greater total distance than WM (9193.21 ± 1261.35 m vs. 6777.92 ± 1506.15 m; *p* = 0.001). Furthermore, their relative distance was significantly higher than that of WM, FB, and CD (114.29 ± 7.67 m/min vs. 111.08 ± 6.86 m/min vs. 107.44 ± 7.92 m/min vs. 96.00 ± 8.31 m/min, respectively; *p* < 0.001). The Player Load for CM was also significantly greater than for CD and WM (945.71 ± 135.82 a.u. vs. 749.78 ± 131.27 a.u. vs. 670.67 ± 157.07 a.u., respectively; *p* = 0.001), as was their distance covered below 20 km/h when compared to WM (8702.93 ± 1271.89 m vs. 6306.58 ± 1404.94 m; *p* = 0.001). Conversely, CD recorded the highest values for maximum speed, which was significantly greater than that of CM (31.78 ± 1.20 km/h vs. 29.07 ± 1.94 km/h; *p* = 0.035). The FB group showed higher values for sprint distance compared to CM (168.11 ± 91.69 m vs. 86.86 ± 55.99 m; *p* = 0.065), indicating a non-significant trend. All data are presented as mean ± standard deviation and are detailed in Table 1.

The supervised multivariate analysis of PLS-DA was applied to discriminate the playing positions of the soccer players in relation to the external load (GPS) variables after the six championship matches, as observed in Figure 2. Despite a significant *p* in the permutation test, the model presented only moderate goodness-of-fit (R^2^) and a low predictive ability (Q^2^). For the sample included in this study, the WM and F playing positions exhibited the greatest variation between matches, failing to achieve a homogeneity that characterizes these groups (Figure 2, left panel). Therefore, we chose to deepen the analysis and repeat the PLS-DA without the two most heterogeneous positional groups, WM and F, considering that the samples from these two groups overlapped with the other positions. Therefore, the comparisons involving WM and F groups were inconclusive for our analysis. The analysis without these most heterogeneous groups may better explain the external load findings according to playing position and their association with the urinary metabolomic profile of the remaining player groups.

This new analysis, which included only the external load (GPS) variables for the CM, FB, and CD groups, from the six championship matches, is described in Figure 3. As expected, the model presented an enhanced R^2^ and Q^2^, improving the feasibility of this analysis. It is notable that the CM group is differentiated from the CD group, and the FB group is differentiated from both the CM and CD groups, although their ellipses still slightly overlap or are in proximity (Figure 3, left panel). Furthermore, the relevance of the external load variables according to playing position can be observed in the loading plot and the estimated variable importance values of the VIP-score. These results show a stronger association between players in the CM group and the variables of relative distance, explosive efforts, player load, total distance, and distance below 20 km/h. The CD group was more strongly associated with maximum speed, sprint distance, and deceleration. Meanwhile, the FB group was less distinctly characterized, presenting an intermediate profile between the other two groups without a single highly weighted variable that distinctly discriminates this group. Conversely, the variables: high-intensity running distance; acceleration; explosive efforts; and deceleration; presented lower VIP-score values, indicating they were less capable of discriminating between the groups (Figure 3, right panel).

### 3.2. Urinary Metabolomic Description and Its Association with External Load by Playing Position

A total of 38 metabolites were identified 24 h post-match, as listed in Table 2 with their respective VIP scores. Among these, based on the group separation observed in the loading plot (Figure 4), ten metabolites (VIP score > 1.0) were identified as the most relevant for the variables studied: methylguanidine, trimethylamine, 4-hydroxyphenylacetic acid, pyruvate, glucose, formate, glycine, dimethylglycine, tyrosine, and uracil. Of these, only trimethylamine, dimethylglycine and uracil presented significant differences in univariate analysis (*p* < 0.05). Subsequently, PLS-DA was performed to discriminate between playing positions in relation to internal load, as represented by the metabolites detected in urine after the six championship matches (Figure 4). The model presented a moderately high goodness-of-fit (R^2^), but a low predictive ability (Q^2^). This indicates that the model is better suited to describing the studied sample set and has limited generalizability to external samples.

The loading plot (Figure 4, right panel) suggests an association of the CM group with uracil, 4-hydroxyphenylacetic acid, methylguanidine, trimethylamine, dimethylglycine, and glucose; the FB group with glucose, tyrosine, and formate; and the CD group with tyrosine, formate, pyruvate, and glycine. These results indicate alterations in the urinary metabolomic profile, involving carbohydrate metabolism and the TCA cycle, amino acid and peptide metabolism, pyrimidine metabolism, and ketone body pathways, which discriminate the post-match metabolic recovery profiles. Thus, the metabolic profiles of the CM group appear to be more closely related to muscle damage and inflammatory markers. In contrast, those of the FB and CD groups are more strongly associated with energy metabolism and oxidative stress.

### 3.3. Integrative Analysis of Internal and External Load Variables

The correlations between the external load variables with urinary metabolites by playing position after the championship matches were performed via Pearson matrix (Appendix A) and a heatmap (Figure 5). Specifically, we found positive correlations between glucose and relative distance (*p* < 0.01); between 4-hydroxyphenylacetate and total distance, explosive efforts, deceleration, and distance below 20 km/h, all with (*p* < 0.05); and between uracil and relative distance (*p* < 0.05). Conversely, negative correlations were observed between trimethylamine and dimethylglycine with maximum speed (*p* < 0.05); and between glucose and acceleration (*p* < 0.01). Appendix A) include an expanded heat map detailing all pairwise correlations between metabolites and external load variables, along with a comprehensive correlation table containing all r-values.

## 4. Discussion

The present study demonstrated that urinary metabolomics could identify differences between positions in professional soccer players, as well as discriminate specific physiological responses associated with the external loads imposed during matches—which is in line with the presented hypothesis. Each position exhibited specific patterns of physical performance and related metabolic profiles: the CM group showed higher values for distance covered at speeds below 20 km/h, total distance, relative distance, and player load, and their profiles were associated with metabolites related to muscle damage and inflammatory markers; the FB group showed greater sprint distance; and the CD group demonstrated higher maximum speed, and were linked to energy metabolism and oxidative stress. The observed alterations involved pathways of carbohydrate metabolism, amino acids, antioxidants, and inflammatory processes, reflecting distinct post-match recovery profiles according to playing position.

Playing position in soccer is a key determinant of the physical demands encountered during match-play. Our findings are supported by the work of Curtis et al. [17], who quantified positional match demands in male soccer and reported that players at the CM position covered substantially greater total distance (9941 ± 2140 m) than defenders (Effect Size [ES] = 0.45 ± 0.41). The same authors noted that, due to the pivotal role of their position in offensive transitions, CMs engage in low- to moderate-intensity activities more frequently and for longer durations than players in other positions. Similarly, a study by Teixeira et al. [3] on youth soccer across three age groups (U15, U17, and U19) found that the highest total distance was covered by players at the CM position (5456.9 ± 1565.9 m), with significant differences observed between players in CD and F positions in high-speed running and sprint distance.

Our data further highlights distinct physical profiles between the CM and CD groups. Specifically, group CM exhibited higher values in relative distance, explosive efforts, player load, total distance, and distance below 20 km/h, whereas the CD group demonstrated superior performance in maximum speed, sprint distance, and deceleration. The partial least squares-discriminant analysis (PLS-DA) multivariate model, which associated playing position with external load across the six championship matches, also differentiated the FB group from both the CM and CD groups (Figure 3, left panel). These observations align with previous evidence indicating lower physical outputs for CDs and higher demands for WMs in high-intensity running and total sprint distance—differences largely attributable to tactical roles and attacking positioning [18]. Due to their specific in-game functions, wide players exhibit greater high-speed running and sprinting volumes than other positions [19]. Furthermore, Modric et al. [20] reported that midfielders performed more accelerations and decelerations than players in other positional roles.

The integration of NMR-based metabolomics analysis with external load metrics, utilizing VIP scores, enabled the assessment of the cumulative influence of each metabolic feature within the PLS-DA model. This approach revealed distinct post-match metabolic profiles present 24 h after championship games, characterized by alterations in key metabolites: methylguanidine, trimethylamine, 4-hydroxyphenylacetate, pyruvate, glucose, formate, glycine, dimethylglycine, tyrosine, and uracil. While this model successfully identified key metabolites associated with match load, its predictive power (Q^2^) is low, and the results are specific to the studied context The metabolite list should therefore be interpreted as representative of the physiological stress from the specific matches analyzed, not as a universal profile for all soccer players. This limited generalizability is an expected constraint, attributable to the myriad biological and contextual variables in soccer. Despite this, the study’s core novelty—the ability to discriminate post-match metabolomic profiles and correlate them with external load—establishes a foundational methodology for future research.

Our results align with previous findings; for instance, Marinho et al. [21] identified several post-game metabolites, including formate, which was suggested to be linked to energy yield through aerobic metabolism. The presence of formate immediately after matches may be explained by the oxidation of branched-chain fatty acids. Methylguanidine and dimethylglycine are metabolites derived from amino acid metabolism and from methylation processes. Methylguanidine has been associated with intense physical stress and creatine metabolism, potentially indicating increased energy demand. Dimethylglycine, conversely, is involved in energy metabolism and the regulation of inflammatory responses, likely reflecting the high physical exertion required during matches [1,22].

The energetic demands imposed by training load on the body occur in response to increased metabolic activity from skeletal muscle contraction. This process utilizes energy metabolites such as carbohydrates and lipids to meet the ATP requirements of muscles [23]. Glucose and fatty acids serve as primary energy sources oxidized by muscles during exercise, with their utilization influenced by muscle glycogen content, diet, exercise intensity, and duration. Pyruvate, a key end-product of glycolysis, enters the tricarboxylic acid (TCA) cycle. Both pyruvate and blood lactate levels typically decrease within one hour after exercise; however, elevated concentrations of these metabolites can still be detected in urine 24 h post-exercise [24]. Exercise induces significant alterations in amino acid metabolic pathways, ATP metabolism, glycolysis, free fatty acid beta-oxidation, and ketone body metabolism [25,26,27]. The presence of specific amino acids in the urinary metabolome may be explained by their contribution to oxidative phosphorylation and their role as substrates for gluconeogenesis, ketogenesis, and protein synthesis—particularly following endurance exercise [24,28]. Supporting our findings, studies employing sports-associated metabolomics have further demonstrated that ketone bodies are generated and amino acids are converted to glucose when carbohydrate availability is limited.

The urinary metabolomic profile, when associated with the external load in soccer players, can serve as a crucial internal load marker for discriminating post-match metabolic recovery profiles. Our results align with recent investigations in this field. For instance, Pellegrino et al. [29] demonstrated that the activation of aerobic metabolic pathways, upregulation of the tricarboxylic acid (TCA) cycle, fatty acid β-oxidation, and amino acid metabolism collectively contribute to lower systemic levels of fatty acids, triglycerides, and cholesterol. This metabolic shift accelerates the utilization of energy substrates and reduces fat accumulation. In agreement with our findings, post-match energy metabolism and recovery can be viewed as a reflection of this activation of metabolic pathways, which facilitates the rapid mobilization of energy substrates following intense physical load and aids in the recovery process. Furthermore, the relationship between inflammatory responses and oxidative stress may be interpreted through this metabolic activation. As highlighted by Pellegrino et al. [29], enhanced aerobic metabolism not only improves energy utilization but can also generate free radicals, leading to oxidative stress. This provides a mechanistic justification for the post-game inflammatory response, arising from both metabolic pathway activation and the production of inflammatory mediators.

The study by Kim et al. [30] raises an important consideration regarding increased trimethylamine excretion, which has been reported in patients with renal disease—a condition characterized by diminished flavin-containing monooxygenase enzyme activity. Although not directly related to soccer, this finding offers a valuable perspective on the broader impact of metabolic activity. It prompts reflection on the relationship between metabolism and renal function in athletes, particularly those subjected to high physical loads. Theoretically, trimethylamine excretion could serve as a marker of altered lipid metabolism or inflammatory response, both of which may be influenced by exercise load.

Metabolomic analysis conducted after soccer matches revealed that CMs exhibited profiles more closely associated with muscle damage and inflammatory markers, likely attributable to the higher physical loads they sustained during matches compared to the FB and CD groups. In contrast, the FB and CD groups demonstrated metabolic profiles linked primarily to energy metabolism and oxidative stress. Elevated levels of muscle damage and inflammatory responses following soccer matches have been previously documented in male athletes [31,32]. These findings are further supported by Mohr et al. [33], who observed a significant effect of muscle damage and inflammation on performance in competitive soccer players. Specifically, the study reported a pronounced reduction in player performance when recovery was limited to three days, due to increased feelings of muscle pain and stiffness. This impaired recovery consequently led to diminished explosive and anaerobic performance during an intensified weekly microcycle featuring three matches.

The observed correlations between GPS-derived external load parameters and the metabolomic profile help elucidate how physical exertion affects distinct physiological processes (Figure 5). Positive correlations between glucose and relative distance (*p* < 0.01) suggest increased energy utilization during prolonged efforts, reflecting heightened energy metabolism. 4-hydroxyphenylacetate—which correlated positively with total distance, explosive efforts, deceleration, and distance below 20 km/h (*p* < 0.05)—may be linked to muscle damage and inflammatory processes, as this metabolite is a tyrosine degradation product known to be altered in response to intense physical stress. Uracil, associated with relative distance (*p* < 0.05), may reflect increased energy metabolism and potential alterations in cellular recovery processes. Conversely, negative correlations between trimethylamine and dimethylglycine with maximum speed (*p* < 0.05), as well as between glucose and acceleration (*p* < 0.01), may indicate impaired high-intensity performance due to inflammatory responses or oxidative stress, given these metabolites’ roles in metabolic pathways that can interfere with peak performance. These associations demonstrate how metabolic profiles are modulated by external load, providing valuable insights into physiological adaptation and recovery mechanisms following physical exertion [24,31,32,33].

The metabolomic changes observed in the study by Vike et al. [34] indicate that the effect of intense prolonged exercise on the human metabolome can persist for up to 24 h post-exertion. Their findings demonstrate increases in lactate, pyruvate, TCA cycle intermediates, nucleotide degradation products, glycerol, fatty acids, acylcarnitines, and ketone bodies following exercise, reflecting ongoing processes of metabolic adaptation and recovery. In contrast, bile acids decreased, and amino acid concentrations changed in divergent directions—a pattern likely explained by their diverse synthetic pathways and multifunctional roles, emphasizing the complexity of post-exercise physiological responses.

Crucially, the persistence of these alterations 24 h after physical exertion underscores the importance of this time point for measurement. It highlights the continued need for metabolic rebalancing, muscular recovery, and regulation of anabolic and catabolic pathways, particularly after exercise that imposes significant energy demands and induces cellular stress. This reference substantiates our methodological choice to analyze urine samples 24 h post-match as a valid and physiologically justified alternative. While future studies may opt for blood sampling at earlier time points, our protocol—utilizing next-day urine collection—was designed accounting for practical logistical constraints involving professional athletes and a high number of matches, which would render repeated blood sampling particularly challenging. Nevertheless, our results must be interpreted with care, since they reflect the metabolomic profile 24 h post-exercise and not immediately post-exercise. Considering this is the first time metabolomics are shown to discriminate among playing positions with different external load during matches on professional soccer players, future studies must take this proof of concept to further use it as a tool for monitoring physiological impact and individualization of recovery in this sport.

Several limitations and potential biases in this study must be acknowledged. First, the timing of urine collection—specifically, the interval between the end of the match and sample acquisition—could influence detected metabolite levels, introducing variability not solely attributable to positional role or physical load. However, the literature lacks a timeline tracking an optimal timeframe for metabolomic responses in urine, our 24 h post-match collection was still able to demonstrate differences between the playing positions as presented here. Future studies should focus on establishing a timeline for these metabolic changes to identify an optimal window for monitoring them. Furthermore, match-specific characteristics such as tactical formation, venue (home or away), scoreline dynamics, and other contextual factors were not controlled for and may represent a source of bias. It is well-established that in-game strategies can significantly influence the external load imposed on different positions. As an observational descriptive study, it was not possible to control these factors; therefore, the results demonstrate associations rather than causation.

The sample size was also relatively small, though data were collected repeatedly over different matches, which may limit the generalizability of the findings. On this topic, there were significant differences in GPS-derived metrics among matches, demonstrating that each match represented a unique physical stressor. Although individual athletes were sampled multiple times, each observation was taken under different match conditions and thus represents a unique response. Limited sample sizes are a common constraint in studies involving elite athletes, given the practical challenges of accessing this population without disrupting competitive schedules. Lastly, inter-individual variations in metabolism may confound the association between match load and identified metabolites. We mitigate some of those confounding factors through standardized team nutrition, sleep protocols, and professional routines, yet unrecognized sources of physiological variation likely remain.

Given the modest sample size, our results should be interpreted as exploratory and hypothesis-generating rather than definitive. Although the repeated-measures design increased robustness, future studies with larger and independent cohorts are needed to confirm these findings and strengthen their applicability to broader soccer populations. Despite these limitations, the results are justified as they provide a better understanding of the framework of metabolomics use and applications in soccer. Given the novelty of this application, a necessary first step is descriptive observation to understand the phenomenon, which was the goal of this study. Future studies may benefit from the results of this descriptive observational design to construct more targeted and powerful experimental clinical trials, involving targeted metabolomics with specific playing positions in professional, high-level athletes.

## 5. Conclusions

The data presented here demonstrates the capability of NMR-based metabolomic analysis performed on urine 24 h post-match to differentiate between the CD, CM, and FB playing positions, but comparisons to WM and F positions were inconclusive. Metabolites with the highest VIP-scores were attributed to the specific physical demands of each position. In this sample, specifically, the metabolic profile of the CM group was more closely associated with muscle damage and inflammatory markers, while the FB and CD groups were linked to energy metabolism and oxidative stress. Considering the low predictive power achieved by the metabolites’ PLS-DA analysis, the results may be specific to the samples studied. Still, metabolomics capacity to identify different patterns between groups of players must be emphasized.

The comparison between external and internal load variables revealed a general pattern of consistency in the metabolic responses across different playing positions, except for the CM group. This study underscores the importance of adopting a personalized approach to monitoring training load and recovery, considering the unique physiological demands inherent to each specific playing position.

## Figures and Tables

**Figure 1 biomedicines-13-02583-f001:**
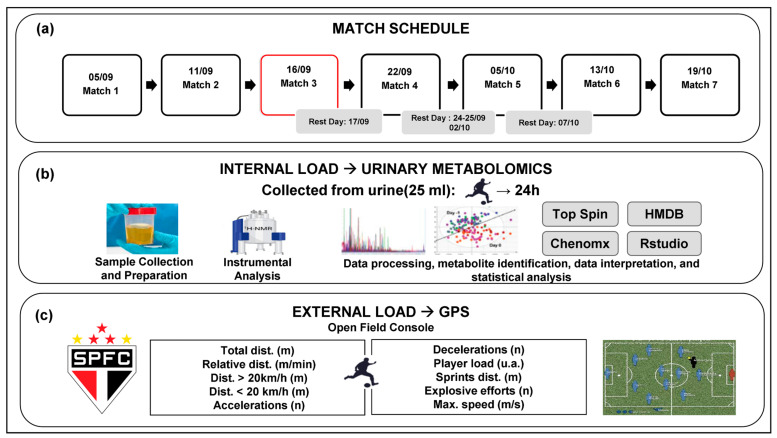
Schematic representation of the study’s methodological design. (**a**) Match schedule for the Qualifying Phases and Semifinal Phase of the São Paulo Under-20 Football Championship (September to October 2022). Daily training sessions and rest periods occurred between matches. Match 3 is highlighted in red, indicating it was excluded from the final sample; (**b**) Workflow for urine sample processing. Samples were collected 24 h post-match for urinary metabolomic analysis via NMR, followed by data processing, analysis, metabolite identification, and interpretation; (**c**) External training load data collection via GPS, aggregated by playing position throughout the season.

**Figure 2 biomedicines-13-02583-f002:**
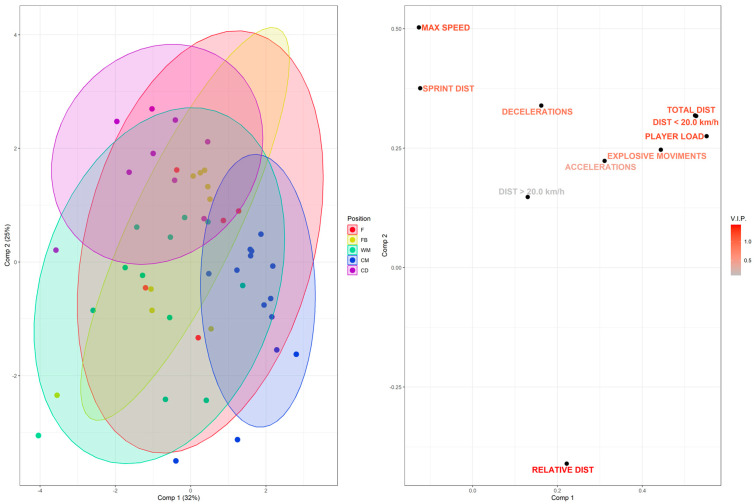
PLS-DA score and loading plots showing the main features that discriminate playing position and external load (GPS) data. Colors represent the different playing positions in the soccer team (red: F; yellow: FB; green: WM; blue: CM; purple: CD), each point represents a player, and triangles represent the group mean, respectively (left panel). Next to it, in the loading plot, are the external load variables and their representations for each playing position, along with a color gradient highlighting the importance of each variable for the projection according to the VIP score (right panel). R^2^ = 0.228; Q^2^ = 0.081; permutation *p* = 0.05.

**Figure 3 biomedicines-13-02583-f003:**
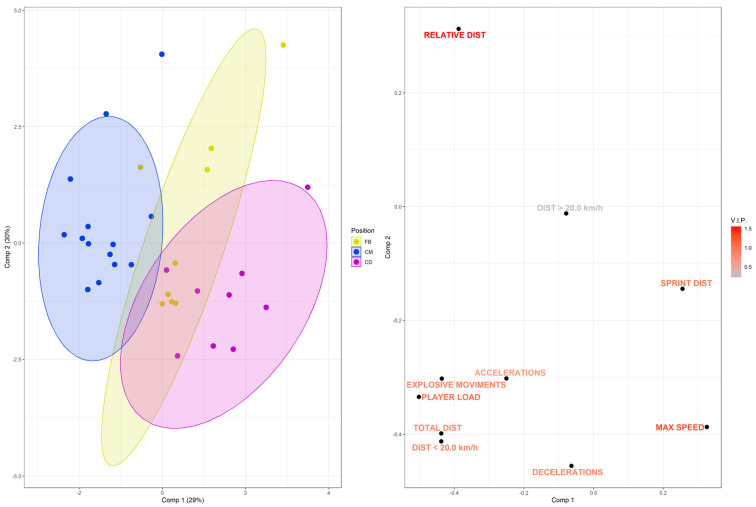
PLS-DA score and loading plots showing the main features that discriminate CM, CD, WB, and external load (GPS) data. Colors represent the different playing positions in the soccer team (blue: CM; yellow: FB; purple: CD), each point represents a player, and triangles represent the group mean, respectively (left panel). Next to it, in the loading plot, are the external load variables and their representations for each playing position, along with a color gradient highlighting the importance of each variable for the projection according to the VIP score (right panel). R^2^ = 0.463; Q^2^ = 0.183; permutation *p* = 0.05.

**Figure 4 biomedicines-13-02583-f004:**
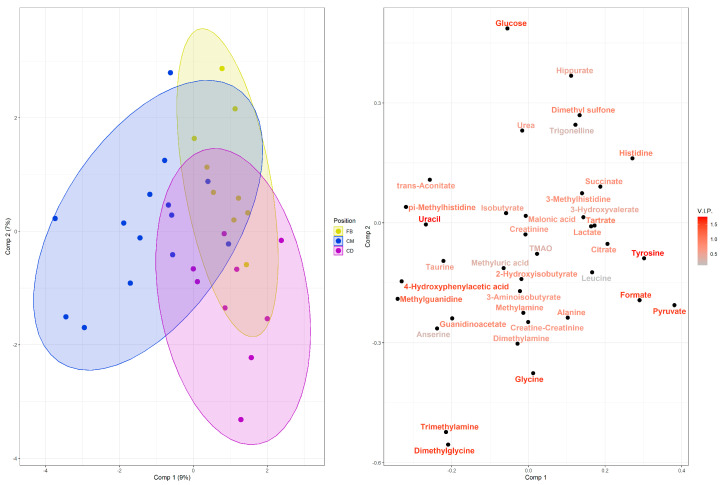
PLS-DA score and loading plots showing the main features that discriminate playing positions, external load data (GPS), and urinary metabolomicss. Colors represent the different playing positions in the soccer team (blue: CM; yellow: FB; purple: CD), each point represents a player, and triangles represent the group mean, respectively (left panel). Next to it, in the loading plot, are the urinary metabolites and their representations for each playing position, along with a color gradient highlighting the importance of each variable for the projection according to the VIP score (right panel). R^2^ = 0.492; Q^2^ = −0.345; permutation *p* = 0.45.

**Figure 5 biomedicines-13-02583-f005:**
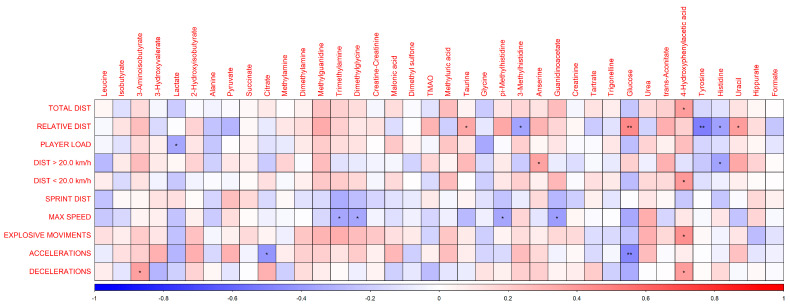
Pearson correlation plot between 38 urinary metabolites and external load variables after championship matches, including positions FB, CM, and CD. Color scales represent positive correlations (red) and negative correlations (blue), and significance is indicated by asterisks: * *p* < 0.05, ** *p* < 0.01.

**Table 1 biomedicines-13-02583-t001:** External Training Load Data by Playing Position during the Six Championship Matches.

External Load (GPS)	Playing Position						
	F (*n* = 5)	FB (*n* = 9)	WM (*n* = 12)	CM (*n* = 14)	CD (*n* = 9)	*p*	1-β
Total Dist (m)	8070 ± 1294.67	8016.67 ± 1719.3	6777.92 ± 1506.15 ^a^	9193.21 ± 1261.35 ^a^	8106.89 ± 1431.50	0.001	0.91
Relative Dist (m/min)	107.40 ± 9.48	107.44 ± 7.92 ^a^	111.08 ± 6.86 ^a^	114.29 ± 7.67 ^a^	96 ± 8.31 ^a^	0.000	0.99
Dist > 20 km/h (m)	635.20 ± 189.55	591.67 ± 255.73	471.33 ± 167.69	490.29 ± 239.62	405.78 ± 78.06	0.220	0.45
Dist < 20 km/h (m)	7434.80 ± 1188.19	7425 ± 1591.40	6306.58 ± 1404.94 ^a^	8702.93 ± 1271.89 ^a^	7701.11 ± 1363.95	0.001	0.94
Player load (u.a.)	861.60 ± 123.98	801.78 ± 188.21	670.67 ± 157.07 ^a^	945.71 ± 135.82 ^a^	749.78 ± 131.27 ^a^	0.001	0.97
Sprints Dist (m)	148.80 ± 54.02	168.11 ± 91.69 ^b^	123.92 ± 72.19	86.86 ± 55.99 ^b^	129.56 ± 30.40	0.065	0.64
Max Speed (km/h)	30.60 ± 1.52	30.33 ± 1.41	30.33 ± 2.19	29.07 ± 1.94 ^c^	31.78 ± 1.20 ^c^	0.035	0.80
Explosive Efforts (*n*)	63.20 ± 20.20	49.56 ± 13.78	49.08 ± 24.07	59.57 ± 14.77	46.67 ± 9.15	0.110	0.42
Accelerations > 2.0 m/s (*n*)	49.20 ± 10.50	41.11 ± 15.68	45.17 ± 14.15	51.36 ± 15.44	45.11 ± 6.55	0.580	0.26
Decelerations < 2.0 m/s (*n*)	41.40 ± 11.50	46.33 ± 14.98	39.92 ± 13.99	44.71 ± 13.53	56.22 ± 19.82	0.230	0.46

F: forward; FB: full-back; WM: wide midfielder; CM: central midfielder; CD: central defender. Values expressed as mean ± standard deviation. *p* = *p*-value (*p* < 0.05). ^a^ difference from the CM group; ^b^ difference from the WB group; ^c^ difference from the CD group; comparison by one-way ANOVA with Tukey’s post hoc test.

**Table 2 biomedicines-13-02583-t002:** Table of VIP scores of urinary metabolites with their respective scores.

Metabolites	VIP	Metabolites	VIP
Uracil	1.75	Succinate	0.79
Tyrosine	1.72	Citrate	0.78
Dimethylglycine	1.58	3-Methylhistidine	0.77
Glycine	1.58	trans-Aconitate	0.73
Formate	1.57	Malonic acid	0.73
Glucose	1.55	3-Aminoisobutyrate	0.69
Pyruvate	1.54	Creatinine	0.69
4-Hydroxyphenilacetic acid	1.46	Creatine-Creatinine	0.68
Trimethylamine	1.44	Dimethylamine	0.65
Methylguanidine	1.24	Urea	0.61
Histidine	0.99	Isobutyrate	0.61
Tartrate	0.98	Taurine	0.59
pi-Methylhistidine	0.96	Hippurate	0.53
Dimethyl sufone	0.94	3-Hydroxyvalerate	0.48
Guanidinoacetate	0.93	Methyluric acid	0.36
2-Hydroxyisobutyrate	0.83	TMAO	0.34
Lactate	0.81	Anserine	0.28
Alanine	0.81	Trigonelline	0.27
Methylamine	0.80	Leucine	0.12

VIP: Variable Importance in Projection.

## Data Availability

All data are within this manuscript. Any other raw data can be obtained upon request to the corresponding author.

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
