# Peer review of "NMR-Based Metabolomics Reveals Position-Specific Signatures Associated with Physical Demands in Professional Soccer Players"

_biomedicines, 2025, doi:10.3390/biomedicines13112583_

Round 1
Reviewer 1 Report
Comments and Suggestions for Authors
Dear Authors,
I want to express my gratitude for the opportunity to revise this manuscript.
The article addresses a very pertinent topic, “how player position influences the metabolomic profile (as a marker of internal load) under known match effort 28 (external load)”.
Below are specific suggestions aiming the manuscript improvement, with line indication.:
The major issues are:
132-156 - Please describe all available information related to the subjects´ characterization. Some examples, training routines, years of experience, competitive level, number of weekly training sessions (specific/strength/injury prevention), and games/competitive events. Another important piece of information is, for example, whether subjects refrain from intense exercise before data collection. Please describe the ethical details. Informed consents fulfilled?
173-205 - Please describe all methodological details, for example, the equipment (manufacturer, version, city, and country), as well as the associated procedures in detail (including time of the season, familiarization, environmental conditions, nutrition, equipment, medicine, warm-up, human resources involved – academic background and experience), preferably with reference support.
206 - Please describe all statistical procedures. For example, the sample power (GPower used?).
P7 – L19-24 – Please consider improving the quality of this section. Some examples include placing the main findings in the first paragraph, including more references, and considering standardization of paragraph size to improve readability (8-12 lines suggested). Finally, please consider presenting suggestions for future research in the last paragraph.
302-319 – The authors´ contributions are normally presented with the name initials; please revise considering the journal template and instructions for authors.
332 - Please double-check the references format, considering the journal template. Some examples, titles in upper and lowercase; the format of the journals; the presentation of the DOI´s.
- - -
The minor issues:
5-23 – Please revise the authors and affiliations format. Please review the journal template and author instructions.
30-32 – Abbreviations seem unnecessary because they are not repeated in the abstract. Please revise.
61-77 – Please consider shorter paragraphs to improve readability. 8-12 lines suggested.
72-74 – Please consider abbreviations.
174-205 – Please consider standardizing the paragraph size.
P7 – L6 – Please consider placing the “p” in italics throughout the manuscript.
P7 – L13 – Please revise the abbreviations throughout the manuscript.
Please revise the document format considering the journal template.
Please revise the English details throughout the manuscript.
Comments on the Quality of English LanguageGlobally, with good quality, but it can be improved.
Author Response
Reviewer 1
Dear Authors,
I want to express my gratitude for the opportunity to revise this manuscript.
Author’s response: Thank you for your revision of our manuscript; we tried to acknowledge your concerns to the best of our ability, and now we think the paper has improved a lot. We included a clean version and a tracked changes version of the manuscript to help you notice the changes made.
The article addresses a very pertinent topic, “how player position influences the metabolomic profile (as a marker of internal load) under known match effort 28 (external load)”.
Below are specific suggestions aiming the manuscript improvement, with line indication.:
The major issues are:
Comment 1: 132-156 - Please describe all available information related to the subjects´ characterization. Some examples, training routines, years of experience, competitive level, number of weekly training sessions (specific/strength/injury prevention), and games/competitive events. Another important piece of information is, for example, whether subjects refrain from intense exercise before data collection. Please describe the ethical details. Informed consents fulfilled?
Author’s response: thanks for the comment, we agree the inclusion of this information will benefit the manuscript. About ethical details, they were inserted in the journal’s platform, but we believe they were not available to the reviewers to ensure blindness of the reviewing process (it contains the ethical board institution name and protocol number, which could identify the authors). But yes, all players fulfilled informed consents and were free to not provide samples for any reason without prejudice against them. We added more characterization information to the topic 2.2, regarding participants. Regarding refraining from exercise, the samples were obtained after the match, which is the phenomenon we are trying to study. The samples were collected in the morning after the match, before any exercise being performed (other than the match itself). In summary:
- Training Routines & Weekly Sessions: During weekly microcycles (7-day periods), the athletes underwent a minimum of 7 training sessions. These sessions were distributed among physical, technical, and tactical training, with an average duration of 70 minutes per session.
- Experience: All included players had at least four years competing at state level, and at least one year at national and international levels, with experience in the U20 and the Senior team.
- Specific / Strength Training: In addition to the group training sessions, some athletes also participated in a minimum of one individual training session (PDI) in the weight room.
- Data Collection During Training/Games: The following data was obtained during daily training sessions and official matches:
- Training time (in minutes)
- Type of training (physical, technical, or tactical)
- Individual training load intensity or Player Load (in arbitrary units, a.u.)
- Distance covered at high intensity (running speed above 20 km/h), in meters
- Number of accelerations and decelerations (in m/s²)
Comment 2: 173-205 - Please describe all methodological details, for example, the equipment (manufacturer, version, city, and country), as well as the associated procedures in detail (including time of the season, familiarization, environmental conditions, nutrition, equipment, medicine, warm-up, human resources involved – academic background and experience), preferably with reference support.
Author’s response: Thanks for the suggestions. We included information about this in topic 2.2, regarding participants, and wrote an even more detailed version of what you asked. Most of this information can be found online, since the club is one of the biggest in Brazil. Specific information were provided by the team’s coaching staff.
The team’s eleven months macrocycle initiated in November 2021, with the preseason and first team competition in January 2022 (São Paulo Junior’s cup), the beginning of the state championship in May, and the national championship in June. In August the team was eliminated from the national championship, and by September 2022 was fo-cused only on the state championship for the data collection period included in this analysis.
All included players had at least four years competing at state level, and at least one year at national and international levels, with experience in the U20 and the Senior team. All included players had at least four years competing at state level, and at least one year at national and international levels, with experience in the U20 and the Senior team.
Regarding human resources involved, the President Laudo Natel Athlete Training Center of São Paulo Futebol Clube consists of a 230,000 m² training area, with one official field and a grandstand for 1,500 people, plus 7 other official fields, 4 social fields, a multi-sports court, 4 dormitories for 110 athletes, 1 dining hall capable of serving 120 people at once, and 1 four-star hotel with a capacity for 148 guests, which was used as a base for the 2014 World Cup in Brazil.
For the period of data collection (fall), all players included in this analysis were housed at the club’s facilities. The center offers 5 daily meals (breakfast, lunch, afternoon snack, dinner, and a late-night snack).
The training center also includes REFFIS (The Center for Sports Physiotherapeutic and Physiological Rehabilitation), comprised of sports doctors, physiotherapists, physiologists, and physical education professionals. Its purpose is to provide specific care for athletes, from injury diagnosis and physiotherapeutic treatments to physical assessments and individualized training programs according to each athlete's needs.
Comment 3: 206 - Please describe all statistical procedures. For example, the sample power (GPower used?).
Author’s response: We did not perform sample power beforehand, since the sample was recruited by convenience. We included information on both topics 2.1 and 2.2, in order to try and clarify this. Volunteers were recruited by convenience, considering the high level of training of the athletes involved in the study. It is not common to have access to professional players during official championship games, so we included as many as possible that fell under the inclusion criteria and out of the exclusion criteria. The team’s roster – one of the top clubs in Brazilian soccer – were composed of forty athletes. However, the twelve included were selected because of their constant participation in games, those considered the core of the team (group one, in their terms). All the included samples were from players that played more than 45 minutes in a given match. The number of matches were all possible from the board’s clearance for data collection to the end of the championship (the team unfortunately lost during the 4th qualifying phase. The project predicted at least three more games until the end of the qualifying phase, and we were not clear to keep up the data collection during the knockout stage if the team were classified). It was a state championship that lasted around 3 months. Despite being a state level championship, it is the major competition for U20 athletes in Brazil, receiving teams and players from the entire country. In addition to the insertion of these details, we calculated the statistical achieved power for the ANOVA comparisons (post-hoc). Out of the ten external load parameters, five of them presented power > 0.8 (four >0.9). Also, for the PLS-DA analysis, the R² and Q² parameters attested the quality of the models. We do acknowledge, however, that the PLS-DA applied to the metabolites (Figure 4) failed to present predictive power and significance in the permutation. Still, the goodness-of-fit was still very relevant (R2 = 0.492), which leads us to believe that for this sample, the separation between groups is real. We added information about this and changed parts of the discussion to account for this limitation. Mainly, the in the discussion now we added: “While this model successfully identified key metabolites associated with match load, its predictive power is specific to the studied context. The metabolite list should therefore be interpreted as representative of the physiological stress from the specific matches analyzed, not as a universal profile for all soccer players. This limited generalizability is an expected constraint, attributable to the myriad biological and contextual variables in soccer. Despite this, the study's core novelty—the ability to discriminate post-match metabolomic profiles and correlate them with external load—establishes a foundational methodology for future research”.
Comment 4: P7 – L19-24 – Please consider improving the quality of this section. Some examples include placing the main findings in the first paragraph, including more references, and considering standardization of paragraph size to improve readability (8-12 lines suggested). Finally, please consider presenting suggestions for future research in the last paragraph.
Author’s response: We are having trouble understanding your criticism in this particular commentary. The discussion already brings the main findings in the first paragraph. We used reference-based arguments 25 times, involving at least 18 papers that were not cited previously in the introduction. All our analysis is followed by an argument from previous evidence from existing literature. We did present suggestions for future studies at the bottom of the discussion – and added another one now, trying to improve the section based on this commentary and others. We acknowledge the paragraphs sizes can be more standardized than they are in the version you reviewed, and now they are lying inside your 8-12 lines suggestion. Can you be more specific if there is any part you think can be improved in this section?
Comment 5: 302-319 – The authors´ contributions are normally presented with the name initials; please revise considering the journal template and instructions for authors.
Author’s response: Thanks for the commentary. We copied and pasted from the submission system, according to the journal orientations.
Comment 6: 332 - Please double-check the references format, considering the journal template. Some examples, titles in upper and lowercase; the format of the journals; the presentation of the DOI´s.
Author’s response: Thanks for mentioning this. The journal’s office formatted the manuscript using their template, and now we think the references are adequate as well.
- - -
The minor issues:
Comment 7: 5-23 – Please revise the authors and affiliations format. Please review the journal template and author instructions.
Author’s response: As I mentioned earlier, the journal’s office formatted the manuscript using their template, so I think now it reads as it should. Sorry for sending it unformatted, but the author’s guidelines say the use of the template is optional upon submission. Now I think we should have formatted the manuscript using the template beforehand.
Comment 8: 30-32 – Abbreviations seem unnecessary because they are not repeated in the abstract. Please revise.
Author’s response: Sorry for this mistake, it was our fault and could be completely avoided. As I will mention in the following responses, in some parts of the abstract the abbreviations for playing positions still reads as they were in Portuguese, our native language. We should have changed them accordingly throughout the text before submitting it. But now the abbreviations you mentioned – the playing positions - are corrected and are used again along the abstract.
Comment 9: 61-77 – Please consider shorter paragraphs to improve readability. 8-12 lines suggested.
Author’s response: We altered the mentioned paragraph and revised the lengths throughout the text.
Comment 10: 72-74 – Please consider abbreviations.
Author’s response: Thanks for the suggestion. In fact, we did use abbreviations later on the paper. Now they are stablished in the introduction when they first appeared, and we reaffirmed them in the methods section for clarity – since there are too many positions, we believe mentioning them only in the introduction can be hard to follow.
Comment 11: 174-205 – Please consider standardizing the paragraph size.
Author’s response: Thanks for the consideration, we reviewed the entire manuscript in order to better standardize the paragraph sizes.
Comment 12: P7 – L6 – Please consider placing the “p” in italics throughout the manuscript.
Author’s response: We revised the text and made this alteration.
Comment 13: P7 – L13 – Please revise the abbreviations throughout the manuscript.
Author’s response: Thanks for noticing this. As I mentioned earlier, the manuscript had a problem with the abbreviations, and now we believe that it has been solved.
Comment 14: Please revise the document format considering the journal template.
Author’s response: As I mentioned earlier, in this version the journal’s office formatted the manuscript using their template, so I think now it reads as it should. Sorry for sending it unformatted, but the author’s guidelines say the use of the template is optional upon submission. Now I think we should have formatted the manuscript using the template beforehand.
Comment 15:Please revise the English details throughout the manuscript.
Author’s response: We performed another thrill English review in the manuscript.
Reviewer 2 Report
Comments and Suggestions for Authors
This ia longitudinal descriptive study that aims to investiagte how the soccer's player position influnces the individual metabolomic profile, under the known external load of effort required as a result of the player position. It an ambitious study with many limitations that can provide important information using the markers of muscle damage and inflamation for personalized training and recovery strategies
In general it is a well organized study and the results and discussion parts are well - written and presented.
There are some points though that he writters should explain/elaborate in order toprovide a better view of their study:
- In the abstract section: lines 30-31 could be re-phrased as to clearly state tht athe "49 individual match observations" refer to the 12 players observed in 6 matches.
- Lines 41-44 The abbreviation MC, LA and ZC are not explained or mentioned anywhre else in the text, please amend
- Materials and methods, please elaborate more on how the number of participants (12) and the number of observed matches (6) were selected, giving if possible in resulting power of the study
- Line 137 -138. Please explain further on the term "free from injury or physiotherapeutic treatment": a) maybe an "and" should be added or is it the one or the other?? b. please state if there was a time frame that the participant was considered free eg for the last year? the last 6 months, never?
Author Response
This is a longitudinal descriptive study that aims to investigate how the soccer's player position influences the individual metabolomic profile, under the known external load of effort required as a result of the player position. It an ambitious study with many limitations that can provide important information using the markers of muscle damage and inflamation for personalized training and recovery strategies
In general it is a well organized study and the results and discussion parts are well - written and presented.
There are some points though that he writters should explain/elaborate in order toprovide a better view of their study:
- In the abstract section: lines 30-31 could be re-phrased as to clearly state tht athe "49 individual match observations" refer to the 12 players observed in 6 matches.
Author’s response: Thanks for your comment. We included a clean version and a tracked changes version of the manuscript to help you notice the changes made. We revised the phrasing to add emphasis. It now reads as: “This was a longitudinal observational descriptive study involving 12 professional soccer players, across six matches during the season, culminating on a total of 49 individual match observations from those players (Central Defenders [CD]=9; Full backs [FB]=9; Central Mid-fielders [CM]=14; Wide Midfielders [WM]=12; Forwards [F]=5).”
- Lines 41-44 The abbreviation MC, LA and ZC are not explained or mentioned anywhre else in the text, please amend
Author’s response: We have to apologize for this one. Those are the abbreviations in Portuguese, and we failed to change it in the English version. Thank you for noticing. We have made the corrections accordingly. Now, MC reads as CM, LA reads as WM, and ZC as CD.
- Materials and methods, please elaborate more on how the number of participants (12) and the number of observed matches (6) were selected, giving if possible in resulting power of the study
Author’s response: Thanks again for the comment. We included information on both topics 2.1 and 2.2 in order to try and clarify this. Volunteers were recruited by convenience, considering the high level of training of the athletes involved in the study. It is not common to have access to professional players during an official championship game, so we included as many as possible that fell under the inclusion criteria and out of the exclusion criteria. The team’s roster – one of the top clubs in Brazilian soccer – were composed of forty athletes. However, the twelve included were selected because of their constant participation in games, those considered the core of the team (group one, in their terms). All the included samples were from players that played more than 45 minutes in a given match. The number of matches was all possible from the board’s clearance for data collection to the end of the championship (the team unfortunately lost during the 4th qualifying phase. The project predicted at least three more games until the end of the qualifying phase, and we were not clear whether to keep up the data collection during the knockout stage if the team were classified. It was a state championship that lasted around 3 months. Despite being a state-level championship, it is the major competition for U20 athletes in Brazil, receiving teams and players from the entire country. In addition to the insertion of these details, we calculated the statistical achieved power for the ANOVA comparisons (post-hoc). Out of the ten external load parameters, five of them presented power > 0.8 (four >0.9). Also, for the PLS-DA analysis, the R² and Q² parameters attested the quality of the models.
- Line 137 -138. Please explain further on the term "free from injury or physiotherapeutic treatment": a) maybe an "and" should be added or is it the one or the other?? b. Please state if there was a time frame that the participant was considered free eg for the last year? the last 6 months, never?
Author’s response: Thanks for the suggestion, we tried to better clarify this matter. For instance, we used the evaluation of the team’s medical staff, and all players cleared for full physical effort and contact, without playing time restrictions, were included in the sample. This part now reads: “being eligible for full physical training and playing time (free from injury or physiotherapeutic treatment by the medical department).” (lines 139-140 of the clean version)
Reviewer 3 Report
Comments and Suggestions for Authors
Notes to Authors
The manuscript describes attempting to identify unique urine metabolomics characteristics based on played positions in “professional” soccer players. The authors do not indicate the tier of professional soccer players utilized nor indicate/isolate positions based on team formation in the abstract, nor say that this will be addressed later. For example, the conditioning and physical expectations on full backs and centre backs changes dramatically from a standard 4-3-3 formation than compared to the currently popular 4-2-3-1 (interestingly this formation is shown in Figure 1 later on), where players have to assume different roles based on ball position. The different formations is mentioned in the introduction, but not the abstract (nor indicated that it would be dealt with later). The variability of “position” is fairly covered in the introduction.
The level of soccer is later mentioned in the Materials and Methods. It may be helpful to briefly mention in the abstract, or at least indicate the information will be provided later in the manuscript etc.
The cohort size is surprisingly small in terms of players and samples. This is mentioned but the statistical impact (i.e. how trust worthy are the results) is not clearly defined.
Taking metabolomics samples only after 24hrs will also presumably limit immediate impact. Not collecting pre- and post-game samples also will confound issues preventing baseline comparisons. While understanding the impacts of even one more sampling for each player for each game can be quite daunting and financially detrimental, it is still needed.
Line 41 the term “MCs” is used but the definitions do not include that position. Did the authors mean central midfielders (CM)?
Line 44 abbreviation “ZCs” is used but not defined
Line 44 abbreviation “LAs” is used but not defined. Left attackers maybe?
Line 45 again MCs without definition. This may be a simple typo.
The details provided are quite good on the cohorts, though fasting before sample collection is not used. Why were fasting samples not collected as this is quite common for exercise related metabolomics?
Team diet is indicated, but not detailed. Presumably all players were consuming similar meals in terms of size and composition. This still leaves a lot of variability which would be expected in the measured results.
Line 182 – what is flash frozen? E.g. liquid nitrogen?
Were any other chemical modifications performed on the samples prior to freezing? E.g. sodium-azide to inhibit bacterial growth, neutralizing pH, etc.?
The NMR details are excellent! Bravo and thank you for providing all the information necessary to evaluate the findings (and to allow others to test/reproduce the findings later etc.). It can be frustratingly rare to find such excellent NMR details in metabolomics papers sometimes.
Regarding the use of Chenomx software,:TSP is (to my knowledge) not one of the standard internal reference compounds (usually something like DSS at a known concentration ~0.5mM etc.). The authors do not mention how the database use was adapted, nor reference why they use TSP and not the IUPAC DSS standard. Other internal reference compounds such as formic acid, imidazole have been attempted, but specifics need to be provided.
Figure 2 – the analysis and display of separation of the extremely small cohort sampling size is not overly convincing. I’m not sure we’re looking at statistically really significant findings.
The small sample size was included in the Discussion section, but the statistical impact nor recommendations on what the readers should interpret from this was not included. This limited range in findings should be directly addressed and defended.
Over all the paper is well thought out and well written. The concept is very interesting. The real hesitation is the extremely small sample size. Perhaps the authors are planning a much larger study and this is a preliminary e.g. proof of concept? This would assure us that more detailed information will be forthcoming?
Author Response
Reviewer 3
Notes to Authors
Comment 1: The manuscript describes attempting to identify unique urine metabolomics characteristics based on played positions in “professional” soccer players. The authors do not indicate the tier of professional soccer players utilized nor indicate/isolate positions based on team formation in the abstract, nor say that this will be addressed later. For example, the conditioning and physical expectations on full backs and centre backs changes dramatically from a standard 4-3-3 formation than compared to the currently popular 4-2-3-1 (interestingly this formation is shown in Figure 1 later on), where players have to assume different roles based on ball position. The different formations is mentioned in the introduction, but not the abstract (nor indicated that it would be dealt with later). The variability of “position” is fairly covered in the introduction.
Author’s response: Thanks for reading our manuscript and for all the detailed comments, critiques and suggestions. Regarding level of the athletes and team’s formation, we added this information now on the abstract and throughout the text. In fact, the initial formation adopted was 4-2-3-1 across all six matches. We agree with you that formation does impact the role of each position, and we should have attested it sooner. Now the manuscript includes this information in the abstract and in topic 2.2.
Comment 2:The level of soccer is later mentioned in the Materials and Methods. It may be helpful to briefly mention in the abstract, or at least indicate the information will be provided later in the manuscript etc.
Author’s response: Thanks for the suggestion. We added the player’s level at the abstract.
Comment 3:The cohort size is surprisingly small in terms of players and samples. This is mentioned but the statistical impact (i.e. how trust worthy are the results) is not clearly defined.
Author’s response:
We thank the reviewer for this valuable observation. We fully acknowledge that our cohort size is relatively small, with twelve players and 49 samples. This limitation is inherent to studies involving professional athletes, as the access to participants is restricted by competitive schedules, training demands, and logistical constraints. Similar metabolomics studies in elite or professional sports have also been conducted with small cohorts, reflecting the practical challenges of this type of research.
To mitigate the potential statistical impact of the limited sample size, we employed a repeated-measures longitudinal design, in which the same athletes were followed across multiple matches. This approach increased the number of observations (n = 49) and allowed us to capture intra-individual variations under different physical load conditions, thereby improving the robustness of the dataset. Additionally, multivariate statistical methods such as PLS-DA were applied with stringent criteria (VIP > 1.0 for urinary metabolites) to reduce the risk of spurious associations. These analytical strategies provide greater reliability to the observed discrimination between playing positions despite the modest cohort size. Further, we added the statistical achieved power for the ANOVA comparisons (post-hoc). Out of the ten external load parameters, five of them presented power > 0.8 (four >0.9). Considering your and other reviewers’ comments, we added considerations about the recruitment of volunteers being by convenience (topics 2.1 and 2.2), because of what we mentioned earlier about the difficulties of achieving large sample sizes when studying high level athletes.
Comment 4: Taking metabolomics samples only after 24hrs will also presumably limit immediate impact. Not collecting pre- and post-game samples also will confound issues preventing baseline comparisons. While understanding the impacts of even one more sampling for each player for each game can be quite daunting and financially detrimental, it is still needed.
Author’s response: We agree with you that 24 hours after the game the metabolomic profile is different than immediately after the match. In this particular data collection, we had no clearance to collect the urine from the players immediately before or after the matches. Still, considering the metabolomic profile is still impacted after 24 hours (Vike et al., 2022; doi: 10.1038/s41598-022-07079-6), and that this is the first time metabolomics are shown to discriminate amongst playing positions with different external load during matches on professional soccer players, we believe the results are of great worth to help develop metabolomics as a tool for understanding soccer impacts on the metabolism. We included a consideration about this at the end of discussion, and had already acknowledge it as a limitation for anyone interested in seeing the metabolomic profile immediately after the match.
Comment 5:Line 41 the term “MCs” is used but the definitions do not include that position. Did the authors mean central midfielders (CM)?
Author’s response: Sorry for this mistake, it was our fault and could be completely avoided. As I will mention in the following responses, in some parts of the abstract the abbreviations for playing positions still reads as they were in Portuguese, our native language. We should have changed them accordingly throughout the text before submitting it. But now the abbreviations you mentioned – the playing positions - are corrected and are used again along the abstract.
Comment 6:Line 44 abbreviation “ZCs” is used but not defined
Author’s response: As mentioned in the earlier response, now this abbreviation reads as “CD”.
Comment 7:Line 44 abbreviation “LAs” is used but not defined. Left attackers maybe?
Author’s response: As mentioned in the earlier response, now this abbreviation reads as “WM”.
Comment 8: Line 45 again MCs without definition. This may be a simple typo.
Author’s response: As mentioned in the earlier response, now this abbreviation reads as “CM”.
Comment 9:The details provided are quite good on the cohorts, though fasting before sample collection is not used. Why were fasting samples not collected as this is quite common for exercise related metabolomics?
Author’s response: This was performed due to the club logistics for data collection. Our staff were only allowed to meet the players during training sessions, so we do not interfere with the team’s usual routine. These are all professional athletes from one of the major clubs here in our country, so we do not have free access to them. Under the circumstances, we managed to collect the urine samples right before the mourning training session on the day following the match. We could ask them to bring a fasting urine sample from the dorm, but we chose to perform the collection before the training session with the orientation of our staff to make sure the samples were adequately collected and not contaminated. Despite agreeing it would be ideal to have fasting samples, and considering they had similar breakfast times between players and within the different time points (matches), we do not believe this would invalidate the results found here.
Comment 10:Team diet is indicated, but not detailed. Presumably all players were consuming similar meals in terms of size and composition. This still leaves a lot of variability which would be expected in the measured results.
Author’s response: The players were under club’s nutrition staff guidance. The meals were prepared in their facilities, and the nutritionist evaluated them regularly to ensure proper macro and micronutrients ingestions were being met. However, we do not had access to the details of their meals. We added this information in topic 2.2.
Comment 11:Line 182 – what is flash frozen? E.g. liquid nitrogen?
Author’s response: sorry for the use of this term, it was a translation problem. We intended to say the samples were immediately frozen. We did not use liquid nitrogen for this data collection.
Comment 12:Were any other chemical modifications performed on the samples prior to freezing? E.g. sodium-azide to inhibit bacterial growth, neutralizing pH, etc.?
Author’s response: No. Other than immediately placing them in -20C temperatures, no other chemical modifications were added to the samples before freezing.
Comment 13:The NMR details are excellent! Bravo and thank you for providing all the information necessary to evaluate the findings (and to allow others to test/reproduce the findings later etc.). It can be frustratingly rare to find such excellent NMR details in metabolomics papers sometimes.
Author’s response: Thanks for mentioning it. Considering the use of metabolomics in soccer is somewhat new, we tried to detail as much as possible the procedures, to ensure future studies could replicate or results. Even for review studies, we believe this kind of information is useful to propose possible sources of variation in results from different cohorts.
Comment 14:Regarding the use of Chenomx software,:TSP is (to my knowledge) not one of the standard internal reference compounds (usually something like DSS at a known concentration ~0.5mM etc.). The authors do not mention how the database use was adapted, nor reference why they use TSP and not the IUPAC DSS standard. Other internal reference compounds such as formic acid, imidazole have been attempted, but specifics need to be provided.
Author’s response:
We thank the reviewer for this important observation. Indeed, DSS is recommended by IUPAC as the standard internal reference compound for quantitative NMR analyses. However, in our study we employed TSP (3-trimethylsilylpropionic acid-d4 sodium salt) as the internal reference. This choice was based on its wide use in NMR-based metabolomics of biofluids, particularly urine, where TSP provides a sharp singlet at 0 ppm, high solubility in aqueous buffer, and minimal overlap with endogenous metabolite resonances. In addition, TSP has been extensively used in urinary metabolomics studies, where it serves primarily as a chemical shift reference. Unlike plasma or serum, where protein binding may interfere, urine is protein-free, making TSP a stable and reliable choice.
Regarding the Chenomx software, it does not require DSS (the IUPAC-recommended standard) as the internal reference. It is possible to specify TSP and insert its concentration (1 mM in our case). From this specification, Chenomx uses the TSP resonance to calibrate the chemical shift axis and adjust the identification of metabolite peaks. In this way, TSP served as the spectral reference for our analyses, and its use is widely accepted in urinary metabolomics studies.
While other internal reference compounds such as DSS, formic acid, or imidazole have been explored, we opted for TSP because of its proven compatibility with urinary samples and its reliability under our acquisition conditions.
Comment 15: Figure 2 – the analysis and display of separation of the extremely small cohort sampling size is not overly convincing. I’m not sure we’re looking at statistically really significant findings.
Author’s response: Yes, particularly in Figure 2, we also believe the separations of the groups were not ideal. We credited this to the wide midfielder and forward playing positions, which had larger external load variations across the studied matches. We now included a phrase emphasizing that with those groups, the separation analysis was inconclusive. This is why we choose to limit this analysis to compare fullbacks, central midfielders and central defenders from this point on. Figures 3 and 4 shows the PLS-DA for external (GPS) and internal (metabolomics) loads of these groups. Despite this, we added R2 and Q2 values for all PLS-DA models. The analysis depicted in Figure 2 had a moderate goodness-of-fit and low predictive power, but a significant permutation p-value (≤0.05). Using only FB, CM and CD groups, the model improved a lot, showing better goodness of fit and even predictive power (still low, but as expected for biological variables). We do acknowledge, however, that the PLS-DA applied to the metabolites failed to present predictive power and significance in the permutation. Still, the goodness-of-fit was still very relevant (R2 = 0.492), which leads us to believe that for this sample, the separation between groups is real. We added information about this and changed parts of the discussion to account for this limitation. Mainly, in the discussion now we added: “While this model successfully identified key metabolites associated with match load, its predictive power (Q2) is low, and the results are specific to the studied context. The metabolite list should therefore be interpreted as representative of the physiological stress from the specific matches analyzed, not as a universal profile for all soccer players. This limited generalizability is an expected constraint, attributable to the myriad biological and contextual variables in soccer. Despite this, the study's core novelty—the ability to discriminate post-match metabolomic profiles and correlate them with external load—establishes a foundational methodology for future research”.
Comment 16: The small sample size was included in the Discussion section, but the statistical impact nor recommendations on what the readers should interpret from this was not included. This limited range in findings should be directly addressed and defended.
Author’s response: We appreciate the reviewer’s insightful comment. The limited sample size was indeed mentioned as a study limitation, but we agree that further clarification is needed regarding its statistical impact and the interpretation of findings. The modest cohort size restricts the generalizability of our results and increases the risk of type II errors. Therefore, the present findings should be interpreted as exploratory, providing preliminary evidence rather than definitive conclusions.
Nevertheless, we emphasize that the repeated-measures longitudinal design (49 samples from 12 athletes across multiple matches) and the use of robust multivariate statistical approaches (PLS-DA with stringent VIP thresholds) mitigate, to some extent, the reduced cohort size by maximizing the amount of information extracted from each participant. This design allowed us to capture intra-individual variability under different match conditions, thereby strengthening the internal validity of the findings.
We have revised the Discussion to directly address this point, explicitly stating that the small sample size limits the external validity of our conclusions, but the observed metabolic signatures by playing position remain consistent and hypothesis-generating. We now also highlight that readers should interpret the results as preliminary evidence supporting the feasibility of NMR-based metabolomics for load monitoring in soccer, while recognizing the need for larger and independent cohorts to validate and extend these findings.
Comment 17 :Over all the paper is well thought out and well written. The concept is very interesting. The real hesitation is the extremely small sample size. Perhaps the authors are planning a much larger study and this is a preliminary e.g. proof of concept? This would assure us that more detailed information will be forthcoming?
Author’s response: As mentioned earlier, we do acknowledge that the sample size is an issue. However, in the context of high-level athletes, it is very difficult to access overly large sample sizes. This is aggravated when studying them during a competition – considering they are all professional athletes and any disturbance in their routine may represent a burden to the outcome of the championship. Still, our data holds value since it demonstrates the possibility of using metabolomics as a marker of internal load in soccer, capable of being integrated to the external load measures. We included several considerations in the discussion to make this more clear, and in the conclusion, now it reads: “Considering the low predictive power achieved by the metabolites’ PLS-DA analysis, the results may be specific to the samples studied. Still, metabolomics capacity to identify different patterns between groups of players must be emphasized”. We hope that the value of these data, even with its limitations, can be recognized and pushes research applying metabolomics to soccer, considering its limited use so far in literature.
Thanks again for your review, we believe you touched on very relevant aspects that really improved the manuscript. You made us rethink the data interpretation and to be more detailed in the assumptions we were making, and now we believe it is more well supported by the presented data.
Round 2
Reviewer 1 Report
Comments and Suggestions for Authors
Dear Authors,
Thank you for considering my suggestions and incorporating them into the manuscript, which has been globally improved. Congratulations.
Below are some specific suggestions with line indications. The manuscript at this point still contains many errors in formatting and some in the text, which require a very careful analysis.
5-8 – Please revise the authors' names considering the journal template and instructions for authors (also standardize the format – e.g. “;” and “,”).
9-22 – Please revise the affiliations format considering the journal template and instructions for authors.
78 – Goalkeeper usually “GK”, please revise throughout the manuscript.
148-150 – Pitch positions are previously in full, in these lines (and after), only the abbreviations are suggested.
153 – Why abbreviate SPFC” in this line and not in line 30? Please consider abbreviating the first appearance in the manuscript, and after that, only presenting the abbreviation.
Figure 1 – Please revise the figure legend format, considering the journal template and instructions for authors.
P8 L7 – Pitch positions are in full, only abbreviations suggested.
P8 L9 – Please consider Player Load “PL” throughout the manuscript.
Figures 2,3 – Please consider improving the quality of the figures.
P11 L63 and figure 3 – Please consider only the “p” in italics, instead of “p-value”.
Table 2 – Please revise the table considering MDPI format.
Figure 5 – Please consider improving the quality of the figure. Again, the “p” should be in italics.
145 – Please change “,” for “.” In the values, aiming for standartization.
154 – “full-back (FB)” / 163 “Variable Importance in Projection (VIP)” – Please correct.
247-253 – “p” in italics. Please revise these kinds of details throughout the manuscript.
330 – Pitch positions in full. Please revise these kinds of details throughout the manuscript.
350-364 – Please consider MDPI format, in this particular case, only authors' initials.
365 – Please correct the funding text.
380 - All references formats should be revised in detail, as they are not according to the journal instructions for authors.
Comments on the Quality of English LanguageCan be improved.
Author Response
Dear Authors,
Thank you for considering my suggestions and incorporating them into the manuscript, which has been globally improved. Congratulations.
Authors’ response: We would like to thank the reviewer for the detailed analysis of our manuscript. We believe that both rounds, one and two, were fundamental for the improvement of a final version of this paper. Again, we added a clean version of the manuscript, and a tracked version with all the changes marked by word’s review tool.
Below are some specific suggestions with line indications. The manuscript at this point still contains many errors in formatting and some in the text, which require a very careful analysis.
5-8 – Please revise the authors' names considering the journal template and instructions for authors (also standardize the format – e.g. “;” and “,”).
Authors’ response: As mentioned before, we had submitted the manuscript originally using the free format submission, as described in the instructions to authors. The editorial office did the kindness to insert the data into the journal’s template. Of course, it is imperative that everything is in accordance to the journal’s norms, so we changed it accordingly. Now, as per instructions for authors, we performed changes to meet the following: “Authors' full first and last names must be provided. The initials of any middle names can be added. The PubMed/MEDLINE standard format is used for affiliations: complete address information including city, zip code, state/province, and country.”. We maintained the email address of the authors, since it was added by the journal’s office.
9-22 – Please revise the affiliations format considering the journal template and instructions for authors.
Authors’ response: We changed accordingly, as mentioned in the last response. Thanks for pointing out.
78 – Goalkeeper usually “GK”, please revise throughout the manuscript.
Authors’ response: Thanks, we changed this and any other use of G for GK when referring to goalkeepers.
148-150 – Pitch positions are previously in full, in these lines (and after), only the abbreviations are suggested.
Authors’ response: We changed the pitch positions for the abbreviations here. In some instances, during the discussion, we chose to maintain the pitch positions in full in order to help with clarity, since there are many abbreviations on the paper. However, some of the authors agree with you that the classical approach of sticking to the abbreviations consistently from beginning to end is more appropriate, so to contemplate your suggestion, we change it throughout the text.
153 – Why abbreviate SPFC” in this line and not in line 30? Please consider abbreviating the first appearance in the manuscript, and after that, only presenting the abbreviation.
Authors’ response: We did not use the team’s name again during the abstract, so felt no need to abbreviate in line 30. We corrected the other time the team’s name was cited throughout the text after line 153 (line 161 of the CLEAN VERSION), using SPFC accordingly.
Figure 1 – Please revise the figure legend format, considering the journal template and instructions for authors.
Authors’ response: We changed it according to the journal template.
P8 L7 – Pitch positions are in full, only abbreviations suggested.
Authors’ response: We changed Pitch positions for abbreviations throughout the text.
P8 L9 – Please consider Player Load “PL” throughout the manuscript.
Authors’ response: We understand Player Load is commonly referred as PL in soccer related literature. However, considering there are already many abbreviations from the playing positions, we feel the manuscript may be heavy on abbreviations. This may make it difficult to understand for a broader audience. Also, since we did not abbreviate other variables of interest (such as distance covered, number of sprints, etc.), it may be prudent to maintain Player Load in full. Also, there is not many times it is repeated throughout the text.
Figures 2,3 – Please consider improving the quality of the figures.
Authors’ response: We changed the outline of figures 2, 3 and 4 to try to attend to your concern. If you still think they need improvement, please be more specific on why. If it is a problem of resolution, we attached all figures in high resolution during the submission process.
P11 L63 and figure 3 – Please consider only the “p” in italics, instead of “p-value”.
Authors’ response: We changed it throughout the text.
Table 2 – Please revise the table considering MDPI format.
Authors’ response: We changed it accordingly.
Figure 5 – Please consider improving the quality of the figure. Again, the “p” should be in italics.
Authors’ response: We changed p to italics, and provided a high resolution figure in the submission process. Also, we changed the presentation of this figure to landscape, to improve visibility.
145 – Please change “,” for “.” In the values, aiming for standartization.
Authors’ response: Are you referring to these values: “… total distance was covered by central midfielders (5,456.9 ± 1,565.9 m), with significant …”? If so, commas are the standard separator for thousands, while points are for decimals. I believe they are accurate throughout the text.
154 – “full-back (FB)” / 163 “Variable Importance in Projection (VIP)” – Please correct.
Authors’ response: We corrected the abbreviations throughout the text.
247-253 – “p” in italics. Please revise these kinds of details throughout the manuscript.
Authors’ response: Thanks for point out, we changed it here and throughout the text.
330 – Pitch positions in full. Please revise these kinds of details throughout the manuscript.
Authors’ response: We corrected the abbreviations throughout the text.
350-364 – Please consider MDPI format, in this particular case, only authors' initials.
Authors’ response: Thanks for pointing this out. We inserted authors’ contributions on the submission system (susy), and the system produced this paragraph. We just copied and pasted it in the submitted version as generated by the system. To answer your concern, we did changed it for initials only.
365 – Please correct the funding text.
Authors’ response: We corrected according to the instructions for authors.
380 - All references formats should be revised in detail, as they are not according to the journal instructions for authors.
Authors’ response: Again, thanks for pointing that out. We used a reference software (Zotero) for constructing the reference list, but in fact it was not configured exactly with the recommended reference format. We now downloaded the entry for the reference format indicated at the instructions for authors.
Reviewer 3 Report
Comments and Suggestions for Authors
The authors have thoroughly and precisely addressed all the reviewer concerns. The detailed responses are excellent and appreciated. The substantial improvements to the clarity of the manuscript are also greatly appreciated. The authors have obviously put a great deal of thought into how to improve the paper, and it shows. The details regarding the use of Chenomx software is especially welcome. There are no further items I can think of that would be needed.
Thank you to the authors for so carefully addressing each concern.
Author Response
Comment 1: The authors have thoroughly and precisely addressed all the reviewer concerns. The detailed responses are excellent and appreciated. The substantial improvements to the clarity of the manuscript are also greatly appreciated. The authors have obviously put a great deal of thought into how to improve the paper, and it shows. The details regarding the use of Chenomx software is especially welcome. There are no further items I can think of that would be needed.
Thank you to the authors for so carefully addressing each concern.
Authors' response:
We would like to thank the reviewer for their many relevant comments. We believe the manuscript has been significantly improved by the feedback and work we did to address the reviewer's concerns. We also appreciate the reviewer's acknowledgment of our effort to respond to their comments to the best of our ability, and for recognizing the value of publishing this data.
However, we note that this positive feedback was sent as a "Round 2" review. As we understand there are no further changes required, could you please verify if any specific action is needed on our end to finalize the process?
Thank you again for the detailed review.
Round 3
Reviewer 1 Report
Comments and Suggestions for Authors
Dear Authors,
Thank you for considering my suggestions and incorporating them into the manuscript, which has been globally improved. Congratulations.
Below are some specific minor suggestions with line indications.
26 – Please describe all abbreviations in full on their first appearance in the manuscript.
191 – Please consider abbreviating metrics.
Figure 1 – Please standardize the upper and lowercase criteria.
Figures 3 and 4 – Please consider not presenting the 2 figures together, aiming to improve readers´ interpretation data.
Table 1 – Please revise the table 1 content and format.
Table 2 – Please revise the table 2 content and format. Please consider a table footnote (for example, with VIP in full).
Figure 3 y-axis with “,” and other values with “.”. Please standardize throughout the manuscript (e.g., also in the discussion section).
328 – Please consider the “p” in italics throughout the manuscript.
354 – Please double-check the references format details.
Please consider improving the English details.
Author Response
Dear Authors,
Thank you for considering my suggestions and incorporating them into the manuscript, which has been globally improved. Congratulations.
Author’s response: Thank you for the detailed review of this manuscript. The paper improved a lot by your comments.
Below are some specific minor suggestions with line indications.
26 – Please describe all abbreviations in full on their first appearance in the manuscript.
Author’s response: Thank you for this suggestion. We have considered spelling out these abbreviations. However, in the field of biochemistry and sports training, abbreviations like NMR (Nuclear Magnetic Resonance) and GPS (Global Positioning System) are considered standard scientific nomenclature and are exceptionally well-known to the specialist readership of this journal. Writing them out in full upon first use can be unnecessary and may even disrupt the reading flow for experts. To maintain conciseness and align with common practice in top-tier journals like Biomedicines, we have retained the abbreviated forms. We believe this enhances readability for our intended audience.
191 – Please consider abbreviating metrics.
Author’s response: Thanks for the suggestion, we incorporated this in the text, according to the abbreviations used in the figures and tables.
Figure 1 – Please standardize the upper and lowercase criteria.
Author’s response: We worked on standardization of the lettering of Figure 1.
Figures 3 and 4 – Please consider not presenting the 2 figures together, aiming to improve readers´ interpretation data.
Author’s response: We changed it accordingly.
Table 1 – Please revise the table 1 content and format.
Author’s response: We updated the formatting and revised the content of table 1.
Table 2 – Please revise the table 2 content and format. Please consider a table footnote (for example, with VIP in full).
Author’s response: We revised the table 2 and included a footnote.
Figure 3 y-axis with “,” and other values with “.”. Please standardize throughout the manuscript (e.g., also in the discussion section).
Author’s response: We changed the figures accordingly. Thanks for pointing out.
328 – Please consider the “p” in italics throughout the manuscript.
Author’s response: We revised all “p” in the manuscript, and now they are all in italics. Sorry for the inconvenience.
354 – Please double-check the references format details.
Author’s response: We revised some journal’s abbreviations that could be updated. Also, one reference had the year of publication two times. We did not find any other inconsistencies on the reference format details. I’m sure the editorial office could help us to revise this after publication.
Please consider improving the English details.
Author’s response: We performed another review of the English grammar, but find little to none errors this time. In the previous opportunity, some grammar choices were improved, and now we believe the English is suited for publication. Again, I’m sure the editorial office would do a revision and minor changes in the English format if that would be necessary. Thanks again for your detailed review.